# How useful are body mass index and history of diabetes in COVID-19 risk stratification?

Sarah-Jeanne Salvy [1]☯*, Geetanjali D. Datta[1]☯, Qihan Yu[1], Marie Lauzon[2], Shehnaz K. Hussain[3], Susan Cheng[4], Joseph E. Ebinger[5], Mark O. Goodarzi[6], Jane C. Figueiredo[1]

1 Cancer Research Center for Health Equity, Cedars-Sinai Medical Center, West Hollywood, CA, United States of America, 2 Samuel Oschin Comprehensive Cancer Institute, Los Angeles, CA, United States of America, 3 Department of Public Health Sciences, UC Davis School of Medicine and Comprehensive Cancer Center, Davis, CA, United States of America, 4 Department of Cardiology, Smidt Heart Institute and Barbra Streisand Women's Heart Center Cedars-Sinai Medical Center, Los Angeles, CA, United States of America, 5 Department of Cardiology and Smidt Heart Institute, Cedars-Sinai Medical Center, Los Angeles, CA, United States of America, 6 Division of Endocrinology, Diabetes and Metabolism, Cedars-Sinai Medical Center, Los Angeles, CA, United States of America

☯ These authors contributed equally to this work.
* sarah.salvy@cshs.org

**Data Availability Statement:** The data that support the findings of this study are available from Cedars-Sinai Medical Center, upon reasonable request. The data are not publicly available due to the contents including information that could compromise

## Abstract

### Objective

This study examines the value of risk stratification by documented diagnosis of diabetes and objectively measured height and weight (BMI) in COVID-19 severity and mortality in a large sample of patients in an urban hospital located in Southern California.

### Methods

Data from a retrospective cohort study of COVID-19 patients treated at Cedars-Sinai Medical Center between March 8, 2020, and January 25, 2021, was analyzed. Sociodemographic characteristics and pre-existing conditions were extracted from electronic medical records. Univariable and multivariable logistic regression models identified associated risk factors, and a regression causal mediation analysis examined the role of diabetes in the association between obesity and illness severity. All analyses were stratified by age (<65 and ≥65).

### Results

Among individuals <65yo, diabetes accounted for 19–30% of the associations between obesity and COVID-19 illness severity. Among patients ≥65yo, having a BMI <18.5 was a risk factor for mortality regardless of diabetes history.

### Conclusion

Our findings have clinical implications in documenting which patients may be at elevated risk for adverse outcomes. More in-depth prospective studies are needed to capture how glycemic regulation may influence prognosis.

research participant privacy/consent. Please direct inquiries to: biodatacore@cshs.org.

**Funding:** The manuscript was partially supported by the National Institute of Diabetes and Digestive and Kidney Diseases (NIDDK; 1R01DK130851) and the Hope Warschaw Center for Integrated Research in Cancer and Lifestyle (Hope Warschaw CIRCL) awarded to SJS, and the by the National NIDDK Eris M. Field Chair in Diabetes Research (P30-DK063491) awarded to MOG. The content is solely the responsibility of the authors and does not necessarily represent the official views of the NIDDK and the Hope Warschaw foundation. The funders had no role in study design, data collection and analysis, decision to publish, or preparation of the manuscript.

**Competing interests:** The authors have declared that no competing interests exist.

## Introduction

Since the onset of the global COVID-19 pandemic, elucidating risk factors of disease severity and mortality have become of primary importance to the medical and public health communities. Several reports have suggested that obesity is associated with COVID-19 progression [1–12], yet it is not entirely clear whether the association between obesity and adverse outcomes is due to excess weight *per se* [11, 13–17], or whether other metabolic conditions oft-associated with obesity are driving these relationships [18–20]. Outside the context of COVID-19, obesity in absence of metabolic abnormalities is not associated with higher risk for all-cause mortality; whereas elevation of even a single metabolic risk factor is associated with increased mortality risk [21]. While emerging studies do support the implication of metabolic and biomarker profiles in COVID-19 severity and such work contributes to our understanding of the mechanisms of disease progression, scientific pursuits and in-depth considerations of cardiometabolic parameters may not be feasible for frontline workers in the context of high-volume critical care for COVID-19 patients. With the constant emergence of new variants and increase in case rates in many locations, already fatigued clinical staff may benefit from heuristics guided by assessment of patient body mass index (BMI) or prior diagnoses of diabetes to guide risk stratification. Utilizing such readily accessible parameters may ease burdens and enable clinical staff to make critical decisions in the face of uncertainty.

Therefore, this study examines the value of risk stratification by BMI and documented diagnosis of diabetes in COVID-19 severity and mortality in a large sample of patients in an urban hospital located in Southern California. Based on previous reports indicating stronger associations between weight, cardiometabolic profiles and COVID-19 severity in younger patients [5–7], and given that age is a significant predictor of mortality [22], all analyses were stratified by age following definitions used in similar studies and health services research [23].

## Materials and methods

### Study sample

The Cedars-Sinai Health System (CSHS) is located in Los Angeles, California with a catchment area of 1·8 million individuals and includes Cedars-Sinai Medical Center (CSMC), Marina Del Rey Hospital (MDRH), and affiliated clinics. In the present study, we included all CSHS patients (1) who received a confirmed diagnosis of COVID-19 infection while being evaluated or treated for suspected COVID-19 between March 8, 2020 and January 25, 2021 and (2) who had objective height and weight measurements and a documented diagnosis of diabetes in their medical record. COVID-19 infection was evaluated using reverse transcriptase polymerase chain reaction of extracted RNA from nasopharyngeal swabs. Until March 21, 2020, patient testing was performed by the Los Angeles Department of Public Health, after which the CSMC Department of Pathology and Laboratory Medicine used the A*STAR FORTITUDE KIT 2·0 COVID-19 Real-Time RT- PCR Test (Accelerate Technologies Pte Ltd, Singapore). For the 3·6% of patients who had COVID-19 testing performed at an outside facility, documentation of a positive test was carefully reviewed by medical staff. Quality control and assurance analyses on data extracted directly from the electronic health record (EHR) were conducted and manual chart review was used to verify collected data where appropriate. Patients provided electronic informed consent via the secure, HIPAA- compliant data capturing tool, REDCap [24, 25]. The CSMC institutional review board approved all protocols for the current study (Proposal # Pro00056865).

## Measurements: Exposures, outcomes and covariates

**Demographics.** Race was self-reported in the EHR as White, African American/Black, Asian, other, and unknown. Ethnicity was self-reported in the EHR and was categorized as Hispanic, non-Hispanic, and unknown. We created a variable combining race and ethnicity into five categories (e.g., Non-Hispanic Black) for use in the statistical models. The participants' zip-codes were linked to census data to provide a measure of area-level income. The low-income category was defined based on zip-code with a median household income below 200% of the federal poverty line. In the absence of other available measures, zip-code level median income was used with the knowledge that caution should be used in its interpretation.

**Smoking status.** Smoking history was ascertained via self-report in the EHR and was categorized as never, current, former, or unknown/not asked.

**Body mass index (BMI).** BMI was calculated from patients' last known objectively assessed height and weight recorded in their EHR, which could have been either prior to- or after COVID-19 diagnoses. To categorize patients based on their calculated BMI, we used the criteria outlined by the World Health Organization, such that: underweight ($<18 \cdot 5$ kg/m$^2$), normal weight ($18 \cdot 5$ to $24 \cdot 9$ kg/m$^2$), overweight ($25 \cdot 0$ to $29 \cdot 9$ kg/m$^2$), and obesity ($\geq 30 \cdot 0$ kg/m$^2$). Consistent with other studies assessing the relationships between weight status and COVID-19, we used normal weight as the reference group.

**Diabetes.** Patients' history of diabetes was coded using the Charlson comorbidity score based on ICD codes documented in the medical record [26].

**Illness severity.** Illness severity was defined in four categories: 1) no hospital admission, 2) requiring any kind of hospital admission, 3) requiring intensive care during hospitalization, and 4) requiring intubation and mechanical ventilation or death. Admission to an intensive care unit (ICU) was identified via time stamps recorded for admission, unit transfers, and discharge. Interventions such as intubation were identified through time stamped orders in the EHR and verified by manual chart review.

**Mortality.** Death was ascertained from time stamps recorded for admission, unit transfers, and discharge.

## Analytic plan

Descriptive analyses were used to assess the distribution of patient demographics and clinical characteristics by mortality status. Median (interquartile range [IQR]) was used to summarize continuous measures, while categorical variables were summarized as counts and percentages. We used Pearson's chi-square test to evaluate statistical differences between patients who were deceased or alive for categorical variables. Bivariate comparisons for continuous variables were evaluated using the Student t-test or Wilcoxon rank sum test. Potential risk factors associated with mortality were identified using univariable and multivariable logistic regression models. Potential risk factors associated with COVID-19 severity were determined using univariable and multivariable ordered logistic regression. Cumulative risk was derived from statistical models including both BMI and diabetes variables with results presented in forest plots.

Because we hypothesized diabetes as a potential mediator on the pathway between obesity and disease outcomes, diabetes was included in the univariate analyses, but not in the multivariable models. The independent contributions of diabetes are however included in the forest plots. We performed regression-based causal mediation analyses to examine the degree to which diabetes mediates the relationship between obesity and our outcomes of interest where one was observed. Causal mediation analysis was conducted using the mediation package in R [27]. The mediate function and 1,000 bootstrap simulations were used to estimate the proportion mediated, and their respective 95% CIs, adjusting for age and sex to account for potential

confounders. A proportion mediated equal or greater than 10% was considered clinically relevant for mortality and severity.

Furthermore, given that age is a significant predictor of mortality, all analyses were stratified by age (<65 and ≥65). The cut-point of 65 years was selected based on definitions used in similar studies and health services research [23]. Analyses were performed using SAS software, version 9·4 (SAS Institute) and R software, version 4·0·3 (R Foundation, Vienna, Austria) with two-sided tests and a significance level of 0·05. The Bonferroni-Holm correction was used to adjust p-values for multiple comparisons.

## Results

### Patient characteristics

In our cohort, 2458 participants were younger than 65 years and 1766 participants were over the age of 65 (Table 1). Mortality in the younger age group was 17·4% (308/1766) and in the older age group was 2·2% (55/2458). Over 40% of individuals over the age of 65 had a history of diabetes and 25% had a BMI >30, while among those younger than 65 years of age approximately 21% had a history of diabetes and 43% had a BMI>30.

### Models

Among those younger than 65 years of age, obesity was associated with increased disease severity relative to normal weight (OR = 1·5, 95%CI: 1·2–1·9; Table 2). Assuming proportional odds, the odds of a severity score = 3 (vs combined scores of 0, 1, 2) was 1·5 time higher among participants with obesity compared to individuals with normal weight. Among participants 65 years of age and older, male (OR = 1·4, 95%CI: 1·1–1·8) and Hispanic (vs non-Hispanic White) individuals were associated with increased odds of disease severity (OR = 1·7, 95%CI: 1·2–2·3; Table 2). Underweight (vs normal weight) was associated with increased odds (OR = 1·9, 95%CI: 1·1–3·1) of mortality.

Patterns for the joint exposure of BMI and diabetes on disease severity varied according to age group. Among those under the age of 65, and relative to patients with normal weight and no history of diabetes, the lowest odds of disease severity were observed among those with a BMI <18.5 and no prior diagnosis of diabetes (OR = 0·4, 95%CI: 0·2–0·7). By contrast, the highest odds of disease severity were observed among individuals with BMI ≥30 and a documented history of diabetes (OR = 3·5, 95%CI: 2·6–4·8; Fig 1A). Among patients 65 years of age or older, an elevate odds of increased disease severity was observed among those with a history of diabetes and a BMI > 30 (OR = 1.5, 95%CI: 1.0–2.1; Fig 1B).

Age-related differences were also observed for the joint association of BMI and diabetes on mortality. Among patients younger than 65, those with diabetes and normal range BMI (OR = 2·6, 95%CI: 1·4–5·0) and those with diabetes and BMI ≥30 (OR = 2·9, 95%CI: 1.1–7·5) were at elevated odds for mortality (Fig 2A). Among patients over the age of 65, elevated mortality was observed among those with BMI <18·5, both among those without (OR = 2·0, 95%CI: 1·2–3·3) and with (OR = 3·1, 95%CI: 1·7–5·7) history of diabetes (Fig 2B). Increased odds of mortality were also observed for those with diabetes and normal BMI range (OR = 1·6, 95%CI: 1·2–2·1), whereas the odds of mortality were slightly lower for patients with overweight without diabetes (OR = 0·69, 95%CI: 0·48–1·00); albeit these latter findings were of borderline statistical significance.

### Causal mediation

In causal mediation models among those younger than 65 years of age, 30% (95%CI:19%-50%) of the association between obesity and disease severity at a level requiring mechanical

**Table 1. Characteristics of patients with objective BMI data, stratified by age.**

| | <65 | | | | ≥65 | | | |
|---|---|---|---|---|---|---|---|---|
| | Alive (N = 2403) | Dead (N = 55) | Total (N = 2458) | p-value | Alive (N = 1458) | Dead (N = 308) | Total (N = 1766) | p-value |
| **Sex** | .. | .. | .. | **0.0004** | .. | .. | .. | 0.11 |
| Female | 1173 (98.9%) | 13 (1.1%) | 1186 (48.3%) | .. | 726 (84.0%) | 138 (16.0%) | 864 (48.9%) | .. |
| Male | 1230 (96.7%) | 42 (3.3%) | 1272 (51.7%) | .. | 732 (81.2%) | 170 (18.8%) | 902 (51.1%) | .. |
| **Ethnicity** | .. | .. | .. | 0.19 | .. | .. | .. | **0.044** |
| Non-Hispanic | 1333 (98.2%) | 24 (1.8%) | 1357 (55.2%) | .. | 1131 (83.1%) | 230 (16.9%) | 1361 (77.1%) | .. |
| Hispanic | 959 (97.3%) | 27 (2.7%) | 986 (40.1%) | .. | 299 (82.4%) | 64 (17.6%) | 363 (20.6%) | .. |
| Unknown | 111 (96.5%) | 4 (3.5%) | 115 (4.7%) | .. | 28 (66.7%) | 14 (33.3%) | 42 (2.4%) | .. |
| **Race** | .. | .. | .. | 0.29 | .. | .. | .. | **0.0046** |
| White | 1367 (97.4%) | 36 (2.6%) | 1403 (57.1%) | .. | 955 (83.4%) | 190 (16.6%) | 1145 (64.8%) | .. |
| African American/Black | 399 (97.3%) | 11 (2.7%) | 410 (16.7%) | .. | 272 (84.7%) | 49 (15.3%) | 321 (18.2%) | .. |
| Asian | 182 (98.9%) | 2 (1.1%) | 184 (7.5%) | .. | 83 (72.8%) | 31 (27.2%) | 114 (6.5%) | .. |
| Other | 332 (99.1%) | 3 (0.9%) | 335 (13.6%) | .. | 123 (83.1%) | 25 (16.9%) | 148 (8.4%) | .. |
| Unknown | 123 (97.6%) | 3 (2.4%) | 126 (5.1%) | .. | 25 (65.8%) | 13 (34.2%) | 38 (2.2%) | .. |
| **Diabetes** | .. | .. | .. | **<0.0001** | .. | .. | .. | **0.010** |
| No | 1914 (98.4%) | 31 (1.6%) | 1945 (79.1%) | .. | 873 (84.5%) | 160 (15.5%) | 1033 (58.5%) | .. |
| Yes | 489 (95.3%) | 24 (4.7%) | 513 (20.9%) | .. | 585 (79.8%) | 148 (20.2%) | 733 (41.5%) | .. |
| **BMI (calculated)** | .. | .. | .. | **0.027** | .. | .. | .. | **<0.0001** |
| N | 2403 | 55 | 2458 | .. | 1458 | 308 | 1766 | .. |
| Mean (SD) | 29.6 (7.5) | 32.1 (8.6) | 29.7 (7.5) | .. | 28.6 (66.2) | 25.0 (6.3) | 28.0 (60.3) | .. |
| Median | 28.8 | 32.1 | 28.8 | .. | 26.2 | 24.2 | 26.0 | .. |
| Q1, Q3 | 24.7, 33.6 | 26.3, 37.1 | 24.7, 33.7 | .. | 22.8, 30.3 | 20.4, 29.0 | 22.2, 30.1 | .. |
| Range | (11.9–74.1) | (18.9–59.5) | (11.9–74.1) | .. | (14.0–2541.7) | (11.4–56.6) | (11.4–2541.7) | .. |
| **BMI category** | .. | .. | .. | 0.13 | .. | .. | .. | **<0.0001** |
| Underweight | 80 (100.0%) | 0 (0.0%) | 80 (3.3%) | .. | 91 (68.9%) | 41 (31.1%) | 132 (7.5%) | .. |
| Normal weight | 554 (97.9%) | 12 (2.1%) | 566 (23.0%) | .. | 519 (80.5%) | 126 (19.5%) | 645 (36.5%) | .. |
| Overweight | 741 (98.4%) | 12 (1.6%) | 753 (30.6%) | .. | 466 (86.0%) | 76 (14.0%) | 542 (30.7%) | .. |
| Obesity | 1028 (97.1%) | 31 (2.9%) | 1059 (43.1%) | .. | 382 (85.5%) | 65 (14.5%) | 447 (25.3%) | .. |
| **Myocardial infarction (Charlson)** | .. | .. | .. | **<0.0001** | .. | .. | .. | **<0.0001** |
| No | 2289 (98.4%) | 38 (1.6%) | 2327 (94.7%) | .. | 1212 (85.2%) | 211 (14.8%) | 1423 (80.6%) | .. |
| Yes | 114 (87.0%) | 17 (13.0%) | 131 (5.3%) | .. | 246 (71.7%) | 97 (28.3%) | 343 (19.4%) | .. |
| **Congestive heart failure** | .. | .. | .. | **<0.0001** | .. | .. | .. | **<0.0001** |
| No | 2201 (98.5%) | 33 (1.5%) | 2234 (90.9%) | .. | 1039 (85.2%) | 180 (14.8%) | 1219 (69.0%) | .. |
| Yes | 202 (90.2%) | 22 (9.8%) | 224 (9.1%) | .. | 419 (76.6%) | 128 (23.4%) | 547 (31.0%) | .. |
| **Chronic pulmonary disease** | .. | .. | .. | 1.00 | .. | .. | .. | 1.00 |
| No | 2034 (97.8%) | 46 (2.2%) | 2080 (84.6%) | .. | 1089 (82.9%) | 225 (17.1%) | 1314 (74.4%) | .. |
| Yes | 369 (97.6%) | 9 (2.4%) | 378 (15.4%) | .. | 369 (81.6%) | 83 (18.4%) | 452 (25.6%) | .. |
| **Chronic pulmonary disease (Charlson)** | .. | .. | .. | 1.00 | .. | .. | .. | 1.00 |
| No | 2036 (97.8%) | 46 (2.2%) | 2082 (84.7%) | .. | 1090 (82.9%) | 225 (17.1%) | 1315 (74.5%) | .. |
| Yes | 367 (97.6%) | 9 (2.4%) | 376 (15.3%) | .. | 368 (81.6%) | 83 (18.4%) | 451 (25.5%) | .. |
| **Admit** | .. | .. | .. | **<0.0001** | .. | .. | .. | **<0.0001** |
| Not Admitted | 1054 (99.9%) | 1 (0.1%) | 1055 (42.9%) | .. | 216 (98.2%) | 4 (1.8%) | 220 (12.5%) | .. |
| Admitted | 1349 (96.2%) | 54 (3.8%) | 1403 (57.1%) | .. | 1242 (80.3%) | 304 (19.7%) | 1546 (87.5%) | .. |
| **ICU** | .. | .. | .. | **<0.0001** | .. | .. | .. | **<0.0001** |
| No ICU | 2172 (99.5%) | 11 (0.5%) | 2183 (88.8%) | .. | 1259 (89.3%) | 151 (10.7%) | 1410 (79.8%) | .. |
| ICU | 231 (84.0%) | 44 (16.0%) | 275 (11.2%) | .. | 199 (55.9%) | 157 (44.1%) | 356 (20.2%) | .. |

*(Continued)*

**Table 1.** (Continued)

| | <65 | | | | ≥65 | | | |
|---|---|---|---|---|---|---|---|---|
| | Alive (N = 2403) | Dead (N = 55) | Total (N = 2458) | p-value | Alive (N = 1458) | Dead (N = 308) | Total (N = 1766) | p-value |
| **Intubated** | .. | .. | .. | <0.0001 | .. | .. | .. | <0.0001 |
| No | 2354 (98.4%) | 38 (1.6%) | 2392 (97.3%) | .. | 1410 (83.9%) | 270 (16.1%) | 1680 (95.1%) | .. |
| Yes | 49 (74.2%) | 17 (25.8%) | 66 (2.7%) | .. | 48 (55.8%) | 38 (44.2%) | 86 (4.9%) | .. |
| **Smoking status** | .. | .. | .. | 0.012 | .. | .. | .. | 0.0002 |
| Current Smoker | 117 (97.5%) | 3 (2.5%) | 120 (4.9%) | .. | 765 (86.5%) | 119 (13.5%) | 884 (50.1%) | .. |
| Former Smoker | 315 (97.2%) | 9 (2.8%) | 324 (13.2%) | .. | 339 (80.3%) | 83 (19.7%) | 422 (23.9%) | .. |
| Never Smoker | 1523 (98.4%) | 24 (1.6%) | 1547 (62.9%) | .. | 321 (77.2%) | 95 (22.8%) | 416 (23.6%) | .. |
| Unknown/not asked | 448 (95.9%) | 19 (4.1%) | 467 (19.0%) | .. | 33 (75.0%) | 11 (25.0%) | 44 (2.5%) | .. |
| **Area-level income** | .. | .. | .. | 0.24 | .. | .. | .. | 0.62 |
| Missing | 23 | 1 | 24 | .. | 17 | 2 | 19 | .. |
| Median income < 200% FPL | 982 (97.2%) | 28 (2.8%) | 1010 (41.5%) | .. | 478 (81.8%) | 106 (18.2%) | 584 (33.4%) | .. |
| Median income > = 200% FPL | 1398 (98.2%) | 26 (1.8%) | 1424 (58.5%) | .. | 963 (82.8%) | 200 (17.2%) | 1163 (66.6%) | .. |

ventilation or death was mediated by diabetes (Table 3). The proportion mediated (PM) was slightly smaller for the other severity outcomes (PMICU = 29%, 95%CI 18%-49% and PMHospitalization = 19%, 95%CI 12%-36%).

## Discussion

In this sample, diabetes accounted for 19–30% of the association between obesity and COVID-19 severity among individuals younger than 65yo. In both age groups, those with both obesity and diabetes had the highest odds of increased disease severity. Among patients older than 65yo, being underweight increased the risk of mortality regardless of diabetes history, whereas overweight seemed to potentially confer a slight survival benefit among patients without a history of diabetes. Our findings raise key clinical questions about age-specific relationships between weight status and COVID-19 outcomes, while documenting which patients may be at elevated risk for hospitalization and adverse outcomes [22].

By design, this study operationalized the mediator of the relationship between weight and COVID-19 outcomes as prior diagnosis of diabetes to distinguish the contribution of pre-existing diabetes from hyperglycemia occurring during hospitalization among COVID-19 patients without prior history of diabetes. In-depth considerations of glucometabolic parameters are often impractical in the context of critical care, and the concordance between historical glucose regulation documented in the medical chart and glycemic control at the time of admission is equivocal. Conceivably, a greater proportion mediated would be observed if the sample was limited to individuals with poorly controlled blood glucose, and prospective metabolic studies are needed to capture variations in glycemic regulation over the course of COVID-19 progression. Nevertheless, our findings are not trivial in indicating that a prior diagnosis of diabetes, even in absence of additional information, accounted for a meaningful proportion of the association between obesity and COVID-19 severity.

Similar to findings in the younger age group, the combination of obesity and diabetes increased the odds of greater illness severity among patients 65 and older. However, the magnitude of the effect was attenuated. Importantly, individuals with a BMI <18.5 were at increased risk of mortality regardless of diabetes history. This finding contrasts with widely publicized relationships between obesity and COVID-19 severity, while consistent with studies documenting age-specific relationships between obesity and COVID-19 outcomes [5, 6, 28].

**Table 2. Logistic models stratified by outcome and age.**

### Ordinal logistic models for illness severity

| | <65 | | | | ≥65 | | | |
|---|---|---|---|---|---|---|---|---|
| | Univariable | | Multivariable | | Univariable | | Multivariable | |
| | OR (95% CI) | p-value | OR (95% CI) | p-value | OR (95% CI) | p-value | OR (95% CI) | p-value |
| **Sex (Male vs. Female)** | 2.2 (1.9, 2.7) | <0.0001 | 2.0 (1.7, 2.5) | <0.0001 | 1.4 (1.1, 1.7) | 0.0010 | 1.4 (1.1, 1.8) | 0.0006 |
| **Race and Ethnicity** | .. | <0.0001 | .. | <0.0001 | .. | 0.0004 | .. | 0.022 |
| Hispanic vs Non-Hispanic White | 1.0 (0.8, 1.2) | 0.95 | 0.8 (0.6, 1.0) | 0.030 | 1.8 (1.3, 2.4) | <0.0001 | 1.7 (1.2, 2.3) | 0.0008 |
| Non-Hispanic Asian vs Non-Hispanic White | 0.4 (0.3, 0.6) | <0.0001 | 0.5 (0.3, 0.7) | <0.0001 | 1.0 (0.6, 1.6) | 0.98 | 1.0 (0.6, 1.6) | 0.96 |
| Non-Hispanic Black vs Non-Hispanic White | 1.0 (0.7, 1.3) | 0.79 | 0.9 (0.7, 1.2) | 0.44 | 1.2 (0.9, 1.6) | 0.42 | 1.3 (0.9, 1.8) | 0.25 |
| Non-Hispanic Other vs Non-Hispanic White | 0.8 (0.5, 1.3) | 0.48 | 0.7 (0.4, 1.1) | 0.12 | 1.3 (0.8, 2.3) | 0.48 | 1.2 (0.7, 2.1) | 0.40 |
| Unknown vs Non-Hispanic White | 0.2 (0.2, 0.4) | <0.0001 | 0.2 (0.1, 0.4) | <0.0001 | 1.9 (1.0, 3.8) | 0.036 | 1.3 (0.6, 2.6) | 0.43 |
| **Diabetes** | 2.8 (2.2, 3.4) | <0.0001 | .. | .. | 1.1 (0.9, 1.4) | 0.31 | .. | .. |
| **BMI category (normal = reference group)** | .. | <0.0001 | .. | <0.0001 | .. | 0.48 | .. | 0.33 |
| Underweight vs normal weight | 0.5 (0.3, 0.9) | 0.016 | 0.3 (0.2, 0.6) | <0.0001 | 1.3 (0.8, 2.0) | 0.24 | 1.3 (0.8, 2.0) | 0.20 |
| Overweight vs normal weight | 1.2 (1.0, 1.5) | 0.17 | 1.0 (0.8, 1.3) | 1.00 | 1.1 (0.8, 1.4) | 0.71 | 1.1 (0.8, 1.4) | 1.00 |
| Obesity vs normal weight | 1.6 (1.2, 1.9) | <0.0001 | 1.5 (1.2, 1.9) | 0.0008 | 1.2 (0.9, 1.6) | 0.21 | 1.2 (0.9, 1.7) | 0.11 |
| **Smoking status (never = reference group)** | .. | <0.0001 | .. | <0.0001 | .. | <0.0001 | .. | <0.0001 |
| Current smoker vs. never | 1.6 (1.1, 2.4) | 0.024 | 1.3 (0.8, 1.9) | 0.47 | 1.4 (0.7, 2.9) | 0.32 | 1.2 (0.6, 2.5) | 0.58 |
| Former smoker vs. never | 1.2 (0.9, 1.6) | 0.21 | 1.1 (0.8, 1.4) | 1.00 | 1.1 (0.8, 1.5) | 0.43 | 1.1 (0.8, 1.4) | 1.00 |
| Unknown/not asked vs. never | 5.5 (4.3, 7.0) | <0.0001 | 5.1 (3.9, 6.5) | <0.0001 | 2.8 (2.1, 3.7) | <0.0001 | 2.7 (2.0, 3.5) | <0.0001 |
| **Median <200% vs ≥200% income** | 1.5 (1.3, 1.8) | <0.0001 | 1.3 (1.1, 1.6) | 0.0016 | 1.1 (0.9, 1.4) | 0.38 | 0.8 (0.6, 1.1) | 0.078 |

### Binary logistic models for death

| | <65 | | | | ≥65 | | | |
|---|---|---|---|---|---|---|---|---|
| | Univariable | | Multivariable | | Univariable | | Multivariable | |
| | OR (95% CI) | p-value | OR (95% CI) | p-value | OR (95% CI) | p-value | OR (95% CI) | p-value |
| **Sex (Male vs. Female)** | 3.1 (1.5, 6.3) | 0.0008 | 2.8 (1.3, 5.9) | 0.0034 | 1.2 (0.9, 1.6) | 0.11 | 1.2 (0.87, 1.6) | 0.23 |
| **Race and ethnicity** | .. | 0.63 | .. | 0.72 | .. | 0.0092 | .. | 0.051 |
| Hispanic vs Non-Hispanic White | 1.6 (0.7, 3.6) | 0.36 | 1.5 (0.6, 3.5) | 0.65 | 1.1 (0.8, 1.6) | 0.60 | 1.1 (0.7, 1.7) | 0.65 |
| Non-Hispanic Asian vs Non-Hispanic White | 0.7 (0.1, 3.7) | 0.58 | 0.4 (0.0, 4.1) | 0.36 | 1.9 (1.2, 3.2) | 0.0088 | 1.7 (1.0, 3.0) | 0.042 |
| Non-Hispanic Black vs Non-Hispanic White | 1.6 (0.6, 4.3) | 0.53 | 1.6 (0.6, 4.2) | 0.65 | 0.9 (0.6, 1.4) | 0.61 | 0.9 (0.6, 1.4) | 0.65 |
| Non-Hispanic Other vs Non-Hispanic White | <0.001(-inf, inf) | 1.00 | <0.001(-inf, inf) | 1.00 | 0.8 (0.4, 1.7) | 1.00 | 0.9 (0.4, 1.8) | 1.00 |
| Unknown vs Non-Hispanic White | 1.8 (0.5, 6.9) | 0.30 | 1.6 (0.4, 6.1) | 0.44 | 2.5 (1.2, 5.2) | 0.012 | 2.2 (1.0, 4.8) | 0.037 |
| **Diabetes** | 3.0 (1.6, 5.6) | 0.0001 | .. | .. | 1.4 (1.0, 1.9) | 0.011 | .. | .. |
| **BMI category (normal as reference group)** | .. | 0.32 | .. | 0.22 | .. | <0.0001 | .. | 0.0004 |
| Underweight vs normal weight | <0.001 (-inf, inf) | 0.98 | <0.001 (-inf, inf) | 0.98 | 1.9 (1.2, 3.0) | 0.0072 | 1.9 (1.1, 3.1) | 0.0086 |
| Overweight vs normal weight | 0.8 (0.3, 1.9) | 0.48 | 0.6 (0.2, 1.5) | 0.20 | 0.7 (0.5, 1.0) | 0.024 | 0.7 (0.5, 1.0) | 0.067 |
| Obesity vs normal weight | 1.4 (0.6, 3.0) | 0.34 | 1.2 (0.6, 2.7) | 0.55 | 0.7 (0.5, 1.0) | 0.066 | 0.8 (0.5, 1.1) | 0.22 |
| **Smoking status (never as reference group)** | .. | 0.016 | .. | 0.054 | .. | 0.0002 | .. | 0.0030 |
| Current smoker vs. never | 1.6 (0.4, 6.5) | 0.43 | 1.4 (0.3, 5.6) | 0.63 | 2.1 (1.0, 4.8) | 0.070 | 2.0 (0.9, 4.6) | 0.14 |
| Former smoker vs. never | 1.8 (0.8, 4.4) | 0.13 | 1.7 (0.7, 4.3) | 0.18 | 1.6 (1.1, 2.2) | 0.0078 | 1.6 (1.1, 2.4) | 0.0046 |
| Unknown/not asked vs. never | 2.7 (1.3, 5.4) | 0.0015 | 2.4 (1.2, 5.0) | 0.0065 | 1.9 (1.4, 2.7) | <0.0001 | 1.7 (1.2, 2.4) | 0.0024 |
| **Median <200% vs ≥200% income** | 1.5 (0.8, 2.8) | 0.24 | 1.2 (0.6, 2.3) | 1.00 | 1.1 (0.8, 1.4) | 0.62 | 1.0 (0.7, 1.5) | 1.00 |

For instance, a study conducted in Southern California found associations between BMI and mortality in COVID-19 patients, particularly among individuals under the age of 60 [8]. In another study conducted in New York City, obesity was associated with respiratory failure but not in-hospital mortality [28]. To our knowledge, our findings linking underweight status and increased mortality risks have not been widely reported in the literature. Underlying frailty or associated diseases are possible explanators of underweight-associated mortality.

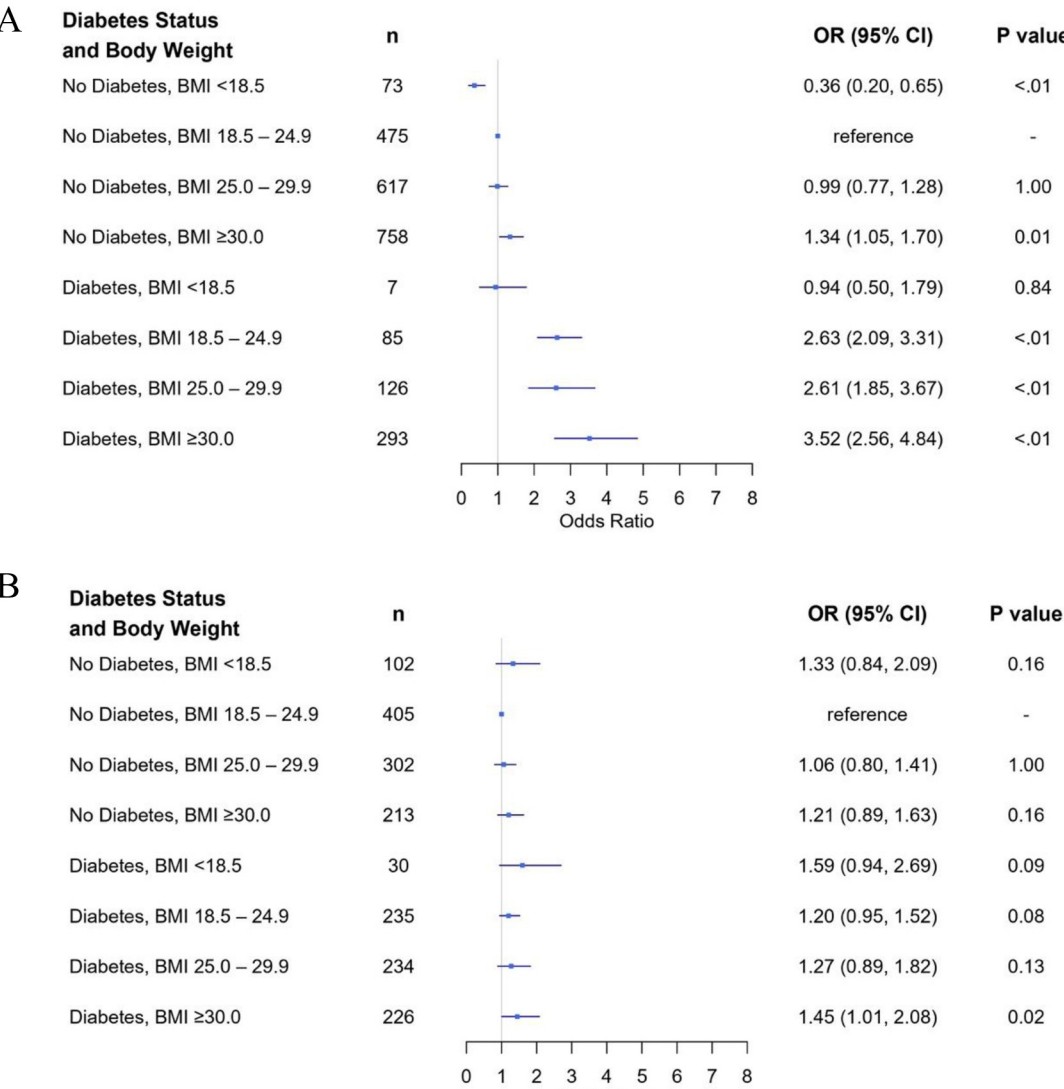

Note: Multivariable regression models are adjusted for sex, race, ethnicity, smoking status and socioeconomic status relative to the federal poverty line.

**Fig 1.** Forest plot of adjusted risk factors for COVID-19 disease severity among patients (A) <65yo and (B) ≥65yo.

The current results are not entirely consistent with previous studies have reported greater odds of poor outcomes among African Americans and Hispanics and those with lower SES [29]. Racial and ethnic minorities, and those with lower SES who seek regular care within the CSHS may not be representative of the larger population in Los Angeles. This explanation is in-line with the null association reported between race and ethnicity and COVID-19 related mortality from an analysis of data from Kaiser Permanente Southern California, an integrated health care organization [8]. Missing data on race and ethnicity might also mask inequalities [30]. Additionally, the limitations of zip-code based measures as proxy for individual-level income are well documented.

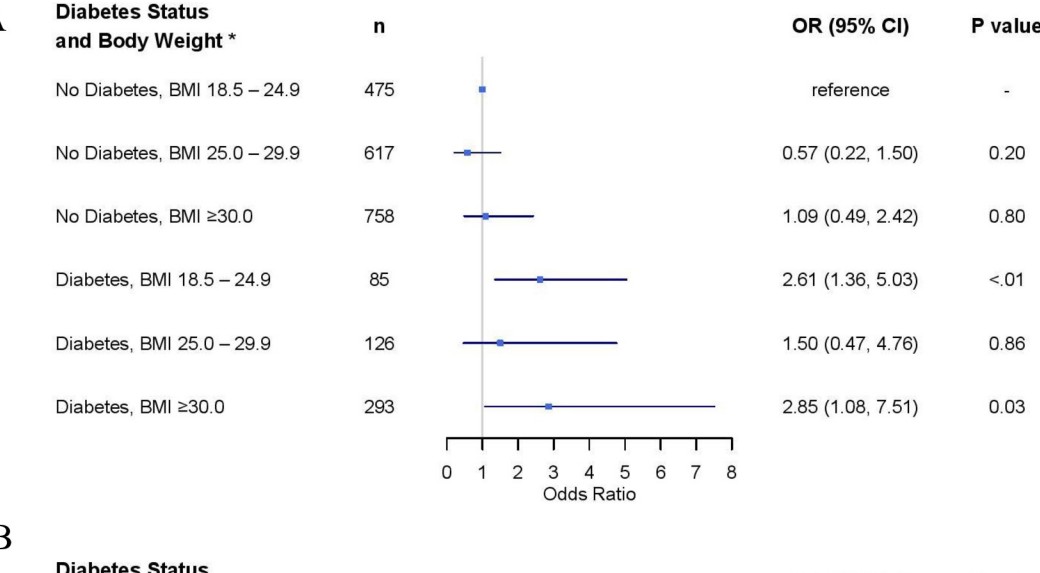

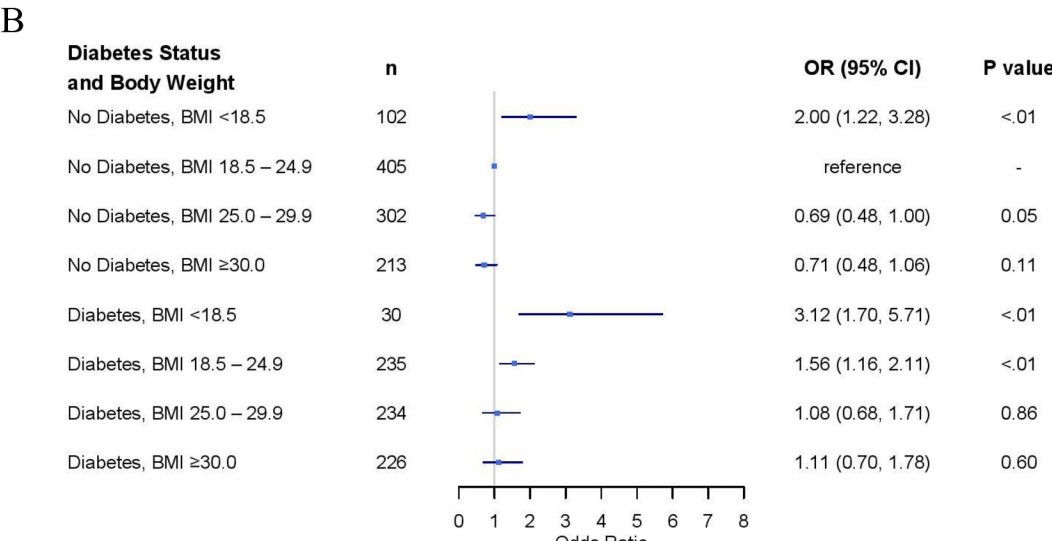

Note: Multivariable regression models are adjusted for sex, race, ethnicity, smoking status and socioeconomic status relative to the federal poverty line.
* No death reported in the population of patients who were <65yo and underweight.

**Fig 2.** Forest plot of adjusted risk factors for COVID-19 mortality among patients (A) <65yo and (B) ≥65yo.

**Table 3. Proportion of the association between obesity and disease severity mediated by diabetes in participants age <65 years.**

| Severity score response | Average proportion mediated (95% CI) |
| --- | --- |
| Not admitted into the hospital | 0.24 (0.15, 0.42) |
| Admitted into the hospital | 0.19 (0.12, 0.36) |
| Admitted into ICU | 0.29 (0.18, 0.49) |
| Mechanical ventilation or death | 0.30 (0.19, 0.50) |

## Limitations

First, it is important to acknowledge that our sample was limited to patients who regularly receive care at a large, urban hospital system. While the availability of objective anthropometric measurements, documented diabetes history and other EHR data increases the rigor of our results, our findings may not be representative of the larger population of individuals with COVID-19 who sought care at a tertiary medical center. Second, history of diabetes and height and weight data were extracted from EHR, however the recency of the data was not considered, thus potentially introducing discrepancy between current and historical data and overall lack of granular specificity of the Charlson comorbidity score used. Finally, the cut-point of 65 years used in the stratification was selected based on definitions used in similar studies [23] and health services research. Conceivably, different cut-offs may reveal different findings.

## Conclusions

This study raises interesting clinical questions about age-specific relationships between weight status, metabolic risk factors and COVID-19 outcomes, while documenting which patients may be at elevated risk for hospitalization and adverse outcomes. Identifying heuristics that clinical staff can utilize to make decisions considering the persistent strain of the COVID-19 pandemic may improve patient outcomes and relieve staff burnout. In the absence of risk markers in the general population, these findings and others highlight the importance of self-management skills and regular medical care among patients with diabetes. COVID-19 has had a disproportionate impact on people with diabetes in challenging adherence and access to diabetes care, thereby potentially resulting in poorer diabetes control and related complications. More in-depth prospective studies are needed to document how glycemic regulation before and after infection influence prognosis.

## Acknowledgments

The authors wish to thank the participants who were included in this analysis.

## Author Contributions

**Conceptualization:** Sarah-Jeanne Salvy, Geetanjali D. Datta, Shehnaz K. Hussain.

**Data curation:** Susan Cheng, Joseph E. Ebinger, Jane C. Figueiredo.

**Formal analysis:** Marie Lauzon.

**Investigation:** Susan Cheng.

**Methodology:** Sarah-Jeanne Salvy, Geetanjali D. Datta.

**Visualization:** Qihan Yu.

**Writing – original draft:** Sarah-Jeanne Salvy, Geetanjali D. Datta, Qihan Yu.

**Writing – review & editing:** Sarah-Jeanne Salvy, Geetanjali D. Datta, Shehnaz K. Hussain, Susan Cheng, Joseph E. Ebinger, Mark O. Goodarzi, Jane C. Figueiredo.

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
