## [Decision Letter · Decision Letter 0]

27 Jan 2022

PONE-D-21-25945

How useful are body mass index and history of diabetes in COVID-19 risk stratification?

PLOS ONE

Dear Dr. Salvy,

Thank you for submitting your manuscript to PLOS ONE. After careful consideration, we feel that it has merit but does not fully meet PLOS ONE’s publication criteria as it currently stands. Therefore, we invite you to submit a revised version of the manuscript that addresses the points raised during the review process.

Please address the following reviewer, additional editor, and journal comments.

We look forward to receiving your revised manuscript.

Kind regards,

Fernando A. Wilson, PhD

Academic Editor

PLOS ONE

https://journals.plos.org/plosone/s/file?id=ba62/PLOSOne_formatting_sample_title_authors_affiliations.pdf”.

“This work is supported by the National Cancer Institute of the National Institutes of Health (1U54CA260591-01). The manuscript was partially supported by the National Cancer Institute (NCI, Grant No. 1R01CA258222; Figueiredo, Salvy & Peterson), the National Institute on Minority Health and Health Disparities Obesity Health Disparities Research Center (NIMHD, Grant No. U54MD0000502; Salvy & Dutton), the Eunice Kennedy Shriver National Institute of Child Health and Human Development (NICHD, Grant No. R01HD092483; Salvy & de la Haye), the Hope Warschaw Center for Integrated Research in Cancer and Lifestyle Award (Hope Warschaw CIRCL, Salvy), and the National Institute of Diabetes and Digestive and Kidney Diseases (NIDDK) Eris M. Field Chair in Diabetes Research (P30-DK063491; Goodarzi). The content is solely the responsibility of the authors and does not necessarily represent the official views of NCI, NIMHD, NICHD, Hope Warschaw CIRCL and NIDDK.”

“This work is supported by the National Cancer Institute of the National Institutes of Health (1U54CA260591-01). The manuscript was partially supported by the National Cancer Institute (NCI, Grant No. 1R01CA258222; Figueiredo, Salvy & Peterson), the National Institute on Minority Health and Health Disparities Obesity Health Disparities Research Center (NIMHD, Grant No. U54MD0000502; Salvy & Dutton), the Eunice Kennedy Shriver National Institute of Child Health and Human Development (NICHD, Grant No. R01HD092483; Salvy & de la Haye), the Hope Warschaw Center for Integrated Research in Cancer and Lifestyle Award (Hope Warschaw CIRCL, Salvy), and the National Institute of Diabetes and Digestive and Kidney Diseases (NIDDK) Eris M. Field Chair in Diabetes Research (P30-DK063491; Goodarzi). The content is solely the responsibility of the authors and does not necessarily represent the official views of NCI, NIMHD, NICHD, Hope Warschaw CIRCL and NIDDK.”

6. Thank you for stating the following in the Funding Section of your manuscript:

“This work is supported by the National Cancer Institute of the National Institutes of Health (1U54CA260591-01). The manuscript was partially supported by the National Cancer Institute (NCI, Grant No. 1R01CA258222; Figueiredo, Salvy & Peterson), the National Institute on Minority Health and Health Disparities Obesity Health Disparities Research Center (NIMHD, Grant No. U54MD0000502; Salvy & Dutton), the Eunice Kennedy Shriver National Institute of Child Health and Human Development (NICHD, Grant No. R01HD092483; Salvy & de la Haye), the Hope Warschaw Center for Integrated Research in Cancer and Lifestyle Award (Hope Warschaw CIRCL, Salvy), and the National Institute of Diabetes and Digestive and Kidney Diseases (NIDDK) Eris M. Field Chair in Diabetes Research (P30-DK063491; Goodarzi). The content is solely the responsibility of the authors and does not necessarily represent the official views of NCI, NIMHD, NICHD, Hope Warschaw CIRCL and NIDDK.”

“This work is supported by the National Cancer Institute of the National Institutes of Health (1U54CA260591-01). The manuscript was partially supported by the National Cancer Institute (NCI, Grant No. 1R01CA258222; Figueiredo, Salvy & Peterson), the National Institute on Minority Health and Health Disparities Obesity Health Disparities Research Center (NIMHD, Grant No. U54MD0000502; Salvy & Dutton), the Eunice Kennedy Shriver National Institute of Child Health and Human Development (NICHD, Grant No. R01HD092483; Salvy & de la Haye), the Hope Warschaw Center for Integrated Research in Cancer and Lifestyle Award (Hope Warschaw CIRCL, Salvy), and the National Institute of Diabetes and Digestive and Kidney Diseases (NIDDK) Eris M. Field Chair in Diabetes Research (P30-DK063491; Goodarzi). The content is solely the responsibility of the authors and does not necessarily represent the official views of NCI, NIMHD, NICHD, Hope Warschaw CIRCL and NIDDK.”

7. We note that you have indicated that data from this study are available upon request. PLOS only allows data to be available upon request if there are legal or ethical restrictions on sharing data publicly. For more information on unacceptable data access restrictions, please see http://journals.plos.org/plosone/s/data-availability#loc-unacceptable-data-access-restrictions.

8. We note that you have included the phrase “data not shown” in your manuscript. Unfortunately, this does not meet our data sharing requirements. PLOS does not permit references to inaccessible data. We require that authors provide all relevant data within the paper, Supporting Information files, or in an acceptable, public repository. Please add a citation to support this phrase or upload the data that corresponds with these findings to a stable repository (such as Figshare or Dryad) and provide and URLs, DOIs, or accession numbers that may be used to access these data. Or, if the data are not a core part of the research being presented in your study, we ask that you remove the phrase that refers to these data.

Additional Editor Comments:

- Please verify and follow PLOS ONE formatting requirements. For example, page and line numbers should be used. Tables must be embedded within the manuscript. Verify formatting of references are correct including use of journal abbreviations. Citations within the text should use square brackets. Please see link below:

https://journals.plos.org/plosone/s/submission-guidelines

Reviewers' comments:

Reviewer's Responses to Questions

**Comments to the Author**

1. Is the manuscript technically sound, and do the data support the conclusions?

Reviewer #1: Yes

Reviewer #2: Yes

2. Has the statistical analysis been performed appropriately and rigorously? 

Reviewer #1: I Don't Know

Reviewer #2: Yes

3. Have the authors made all data underlying the findings in their manuscript fully available?

Reviewer #1: No

Reviewer #2: Yes

4. Is the manuscript presented in an intelligible fashion and written in standard English?

Reviewer #1: Yes

Reviewer #2: Yes

5. Review Comments to the Author

Reviewer #1: This manuscript addresses an important issue and presents novel and interesting results. The manuscript is clearly written, and the major limitations are acknowledge. Unfortunately, my version of the manuscript did not include any tables, despite the fact that 2 tables were referenced in the notes section. There are significant omissions of data, I suspect largely due to the missing tables.

Specific points:

- Please provide a summary of the demographic and clinical data of the sample. This is important to evaluate the generalizability of the results.

-Although measures race/ethnicity, income/socioeconomic status, and smoking history are mentioned in the methods, I see no mention of them anywhere else in the results. Please comment on the influence of these factors on disease severity and mortality in your sample.

Reviewer #2: This study evaluated patients treated at a single medical center for COVID-19 in a retrospective manner looking at how BMI, diabetes, and age influence COVID-19 related outcomes including hospital admission, ICU admission, mechanical ventilation, and death.

They found that up to 30% of obesity related morbidity/mortality in COVID-19 is due to diabetes. In addition, they found that patients with age >65 years who are underweight have a higher odds ratio for adverse outcomes in COVID-19 whereas those age >65 with BMI >30 may have a lower odds ratio for adverse events.

My opinion overall is that this study asks an important question (what is the interaction between obesity and DM in COVID19 outcomes), has sound methodology, draws results and conclusions supported by the presented data, and reasonably addresses limitations. However, the writing needs refining to be more clear, references need to be more appropriately used, and tables 1 and 2 need to be presented for review.

I favor accepting this manuscript with these minor changes.

Commentary:

Introduction:

Overall the introduction needs to be re-worked to discuss the current literature on the relationship between COVID-19 outcomes, obesity, age, and diabetes more clearly as in its current form it reads ambiguously.

Paragraph 1 – reference 1 does not seem to be relevant as it is a commentary without primary data and does not address the question of obesity as the sentence it is cited in does – I would consider changing this sentence so the reference is relevant or removing it altogether

Paragraph 1 – reference 12 does not discuss obesity, only discussing fasting blood glucose – it may be better used elsewhere

Paragraph 1 – sentence 2 – the authors mention hyperlipidemia but this disease is not mentioned elsewhere in the manuscript at all and does not make sense in the context of the paragraph. I wonder if this was meant to state increased BMI instead. If it was referring to hyperlipidemia specifically, it should be further expounded upon.

Paragraph 1 – reference 13 – the content of this reference (diabetes and lung function) is not explicitly discussed in the paragraph – may consider either adding content or removing this reference.

Paragraph 1 – reference 17 – this reference does not seem relevant as it is related to fasting glucose and pancreatic cancer, is a negative study, and does not mention COVID-19, and is not specifically brought up in the manuscript text

Paragraph 1 – reference 17 – the content of this reference (obesity and lung function) is not explicitly discussed in the paragraph – may consider either adding content or removing this reference.

Paragraph 1 – reference 19 – the content of this reference (pulmonary thrombosis in COVID-19) is not discussed - may consider either adding content or removing this reference.

Paragraph 1 – The final two sentences regarding the difference between scientific pursuits and clinical care are of unclear utility – I am not sure what the point of these are in context of the paper especially since these concepts are never referred to again in the manuscript.

Methods:

Overall these are largely well written and easy to follow.

Demographics – the authors state ‘median household income 200% below the federal poverty line’ – I believe what is meant is ‘median household income below 200% of the federal poverty line’

Body Mass Index – it is stated that last known height and weight were used – authors should clarify if these are prior to COVID diagnosis or after.

Body Mass Index – reference 10 doesn’t need to be cited as using baseline as a comparator group is standard

Analytic Plan – reference 25 does not specify why cut point of 65 is used, only that is frequently used.

Results:

Overall, results section is somewhat confusing to follow, going from age <65 cohort and causal mediation to age >65 cohort and mortality associations, then looking at the interactions between BMI and diabetes by age. It may be clearer to separate this section into patient characteristics first, then univariate associations, then multivariate associations (interactions between BMI, diabetes, and age), then finally causal mediation. This may help the important findings of the paper read more clearly.

Table 1 and table 2 were not available for my review and should be made available.

Discussion:

Overall, the discussion section is fairly clear and easy to understand, and is supported by the data presented.

Limitations:

The second paragraph of the limitations as they discuss racial/ethnic disparities is not really part of limitations and would fit better in the discussion section.

Conclusions:

Overall, the conclusion is supported by the data presented and makes reasonable inferences about need for diabetes care in the general population in the setting of COVID-19.

FIGURE 1 AND 2

None of the strata have an OR of 1 – which group is the reference group in this setting that the rest are being compared to?

6. PLOS authors have the option to publish the peer review history of their article (what does this mean?). If published, this will include your full peer review and any attached files.

Reviewer #1: No

Reviewer #2: No

---

## [Author Response · Author response to Decision Letter 0]

3 Feb 2022

Please see attached "Response to Reviewers" document for point by point responses to reviewer and editor comments.

---

## [Editor Report · Decision Letter 1]

3 Mar 2022

How useful are body mass index and history of diabetes in COVID-19 risk stratification?

PONE-D-21-25945R1

Dear Dr. Salvy,

We’re pleased to inform you that your manuscript has been judged scientifically suitable for publication and will be formally accepted for publication once it meets all outstanding technical requirements.

Kind regards,

Fernando A. Wilson, PhD

Academic Editor

PLOS ONE